# COVID-19 and Kidney Transplantation: Epidemiology, Histopathological Presentation, Clinical Presentation and Outcomes, and Therapeutic Strategies

**Maurizio Salvadori [1],\* and Aris Tsalouchos [2]**

[1]  Department of Renal Transplantation, Careggi University Hospital, Viale Pieraccini 18, 50139 Florence, Italy
[2]  Division of Nephrology, Santa Maria Annunziata Hospital, Via Antella 58, 50012 Florence, Italy;
    aris.tsalouchos@gmail.com
\*  Correspondence: maurizio.salvadori1@gmail.com; Tel./Fax: +39-055-59-7151

**Abstract:** Multiple case series of kidney transplant recipients with COVID-19 have shown increased mortality compared to nontransplant patients. To date, we do not have high-level evidence to inform immunosuppression minimization strategies in infected transplant recipients. Most centers, however, have adopted an early antimetabolite withdrawal in addition to other interventions. The epidemiological problem concerns also dialysis patients and waitlisted patients who have a higher COVID-19 infection diffusion with respect to kidney transplant recipients. Several factors influence mortality among kidney transplant recipients. Among these factors are the age, race, and comorbidity factors, such as hypertension, diabetes mellitus, obesity, and previous respiratory problems. Treatment is still limited. The only effective antiviral drug is remdesivir that should be administered before the development of the cytokine storm. Vaccination seems to be useful, but due to the concomitant immunosuppression limiting its efficacy, at least three or four doses should be administered.

**Keywords:** COVID-19 infection; kidney transplantation; outcomes; immunosuppression; treatment

## 1. Introduction

More and more data highlight the impact of COVID-19 infection in solid organ transplant recipients [1].

The aim of this review is to summarize the existing data on kidney transplant (KTx) recipients, highlighting the epidemiological aspects, the renal histopathology, the clinical presentation and outcomes, and the therapeutic strategies.

## 2. Epidemiology

National registries as well as multicenter studies are providing valuable information on COVID-19 epidemiology in kidney transplant recipients [2]. A European study conducted by ERA-EDTA, analyzing French and Spanish registries, reported an incidence rate of COVID-19 as high as 14/1000 [3]; in particular, the Spanish registry reported a rate of 17.7/1000 [4].

Multicenter studies studying the incidence of COVID-19 either in renal patients and in the general population have been conducted in the USA, Italy, and Spain [5,6]. Such studies reported similar incidences as 2–5 times higher in renal transplants than in the general population. In addition, single-center analyses conducted in countries with a high incidence of SARS-CoV-2 infection in the general population reported a higher incidence in renal transplant patients as well [7–9]. Marinaki et al. [10] performed a systematic literature review including 63 articles published in 2020. Among these studies, the mean age of affected patients is approximately 55 years with a male prevalence and with more deceased donors (74%). Most patients had one or more comorbidities.

According to a Belgium study [11], the SARS-CoV-2 infection was higher in patients on hemodialysis (5.31%) than in transplant recipients (1.4%) and the general population (0.64%) (Table 1).

**Table 1.** SARS-CoV-2 infections in adult patients on hemodialysis, kidney transplant recipients, and the general population observed in Belgium [11].

| Population | General Population | Patients on Hemodialysis % (95%CI) vs. Population | | Kidney Transplant % (95%CI) vs. Population | |
|---|---|---|---|---|---|
| Total | 0.64% | 2.54 (2.23–2.89) | *p* < 0.001 | 1,60 (1.18–2.11) | *p* < 0.001 |
| Men | 0.48% | 1.98 (1.65–2.35) | *p* < 0.001 | 1.24 (0.82–1.78) | *p* < 0.001 |
| Women | 0.78% | 3.21 (2.63–3.86) | *p* < 0.001 | 2.00 (1.26–3.04) | *p* < 0.001 |
| Men vs. women | *p* < 0.001 | *p* = 0.03 | | *p* = 0.55 | |

Similarly, in an England study [12], the SARS-CoV-2 incidence in KTx patients was lower than the hemodialysis patients and waitlisted patients (1.5% vs. 9.6% vs. 4.7%).

In a French study [13], the COVID-19 disease was higher in KTx patients than in the general population (5% vs. 0.3%). The same study identified as risk cofactors obesity, chronic pulmonary disease, and diabetes (Table 2). Finally, a large and well-conducted study based on the UK transplant registry comparing waitlisted patients vs transplant recipients found a higher positive testing for SARS-CoV-2 in waitlisted patients than in transplant recipients (3.8% vs. 1.3%) [14].

**Table 2.** Baseline factors associated with higher incidence of COVID-19 disease in kidney transplant patients according to the French study [13].

| Factors Associated with COVID-19 Disease: Multivariate Model | Odds Ratio (95% CI) | *p* Value |
|---|---|---|
| **Ethnicity** | | |
| **White** | **1** | |
| **Nonwhite** | **2.17 (1.23–3.78)** | **0.007** |
| **Cardiovascular disease** | | |
| **No** | **1** | |
| **Yes** | **0.20 (0.03–1.50)** | **0.12** |
| **Obesity (BMI ≥ 30)** | | |
| **No** | **1** | |
| **Yes** | **2.19 (1.19–4.05)** | **0.01** |
| **Asthma and chronic pulmonary disease** | | |
| **No** | **1** | |
| **Yes** | **3.09 (1.49–6.41)** | **0.002** |
| **Diabetes** | | |
| **No** | **1** | |
| **Yes** | **3.33 (1.92–5.77)** | **<0.001** |

In a very recent study, Mahalingasivam et al. [15] critically examined the SARS-CoV-2 epidemiology in renal patients. The authors highlighted that several studies found a higher incidence of SARS-CoV-2 infection among KTx recipients than in the general population. This fact could be ascribed either to a consequence of the immunosuppression or to a greater access to testing occurring in patients with a higher risk of severe disease.

Studies from regional or national registries, such as England, Belgium, France, and other countries, documented a higher incidence of SARS-CoV-2 infection in patients on dialysis or in patients on a waiting list for kidney transplantation than in transplant recipients [16,17]. This finding may be due to the difficulty of isolating patients on dialysis or to the improving of testing for infection control in such patients. It should also be highlighted that the seroconversion studies have included few nonhospitalized KTx recipients [18,19].

## 3. Renal Histopathology

Several authors observing KTx patients affected by COVID-19 infections [20–30] have described different aspects. The authors are still debating whether the lesions are to be ascribed directly to the virus or if the virus acts indirectly. Several observations found the viral RNA in the area site of the lesions.

Kudose et al. [20] described one case of T cell-mediated rejection. The patient presented with AKI and with hypertension and obesity as the principal risk factors. Three cases of antibody-mediated rejection have been described [21,22] All the latter patients presented with acute kidney injury. Hypertension and diabetes mellitus were the main risk factors.

Acute tubular injury is a frequent clinical manifestation in the above-described patients as well as in others [20,21,23]. Jesperson et al. [24] described a patient with thrombotic microangiopaty (TMA) with hemolytic anemia and deteriorating renal function.

A frequent histopathological lesion is focal segmental glomerulosclerosis (FSGS). Two cases of collapsing glomerulopathy have been described in patients with severe acute tubular necrosis [25,26].

In one of these patients, viral RNA was observed by in situ hybridization. This fact is worth highlighting because it could be proof of a direct action of SARS-CoV-2 in the pathogenesis.

Two other patients presented with a recurrence of FSGS [27,28]. Both recurrences occurred early post-transplantation.

Yamada et al. [29] described a case presenting with AKI associated with a biopsy showing minimal change disease.

Finally, Werthoff et al. [30] described a patient with AKI related to acute tubular necrosis. Werthoff also documented SARS-CoV-2 RNA principally in the tubular cells. Such a kidney infiltration by the virus is again proof of a direct injury of the virus itself, suggesting the viral implication in the lesions observed a possible alternative pathophysiological mechanism.

It should be highlighted that almost all patients described resolved without need of dialysis.

## 4. Clinical Presentation and Outcomes

In a large, already mentioned study [10], clinical data from 420 KTx recipients with a confirmed COVID-19 infection were reported.

The majority of the studies cited in the study observed that the symptoms at presentation are similar to those presented by the general population with fever, dry cough, myalgia, dyspnea, and a flu-like syndrome [31,32]. In a relevant number of transplanted patients, mild or atypical initial presentation has been reported [33–35], and this fact suggests a need for increased vigilance. The illness severity among KTx recipients varies, but several reports describe acute respiratory distress with a severe clinical deterioration in a short period after the onset of the disease [33,36,37].

In an early period, several authors thought that the immunosuppressive status could impair the cytokine release and, as a consequence, determine a milder course of the disease [38]. However, the current literature evidence confuted this theory, and to date, it is well known that KTx recipients acquiring a COVID-19 infection are at high risk principally for the immunocompromised status.

An Italian study reported similar clinical data [39]. In this study, 87% of patients experienced a radiological lung progression, and 73% required escalation of oxygen therapy.

The outcomes of 144 hospitalized KTx recipients in 12 centers in the USA, Italy, and Spain were reported by the TANGO international consortium [40]. AKI occurred in 52% of the cases, and the need for respiratory assistance by intubation occurred in 29%. The TANGO study observed also the main risk factors associated with mortality. At the univariate model, older age, occurrence of diarrhea, and need for higher respiratory rate and higher lactate dehydrogenase, procalcitonin, and IL-6 levels were associated with mortality. At the multivariate model, significant risk factors remained older age, higher respiratory rate, high IL-6, and low estimated filtration glomerular rate.

The France registry reported 279 KTx patients with COVID-19; a relevant number of the hospitalized patients required intensive care unit admission; these patients had a high mortality rate. In this registry, as reported by other studies, independent risk factors for mortality included ≥60 years of age, cardiovascular disease, and respiratory failure [41].

The European Renal Association COVID-19 database found a mortality rate of 21% in KTx recipients with COVID-19; in this database, the mortality rate is higher in patients admitted to the intensive care unit and lower in KTx recipients who did not required hospitalization [42].

Overall, from all these studies, it could be observed that AKI is a common renal presentation in patients with COVID-19, and when associated with the need for intensive care unit admission, it is associated with a higher mortality rate [43,44]. The AKI pathogenesis is multifactorial and includes renal hypoperfusion, multiorgan failure, and cytokine storm.

Whether the incidence of AKI is higher in KTx recipients compared to the general population is still a matter of debate [45,46].

A recent study on COVID-19 in all types of organ transplantation [47] revealed that KTx recipients are at a higher risk of acquiring COVID-19 (OR = 1.92) with respect to recipients of different organs. Similarly, in the study from John Hopkins [48], all major adverse clinical outcomes related to COVID-19 were more prevalent in KTx recipients than in recipients of different organs. In the study of Marinaki [10], the authors found that 30% of KTx recipients needed mechanical ventilation, either invasive (IMV) or noninvsasive (NIV). The same study found a higher rate of AKI in KTx patients, compared to severely ill COVID-19-infected patients in the general population [49]. In the Marinaki study, a substantial proportion of patients who developed AKI, needed renal replacement therapy (RRT).

AKI etiology is discussed. Earlier studies [50] reported direct damage of the proximal tubular epithelial cells by SARS-CoV-2. To date, a cytokine storm is the AKI cause considered as the more probable one [51].

## 5. Prophylaxis and Treatment of COVID-19 Infection in Kidney Transplant Patients

### 5.1. Prophylaxis

Several reports [52] document the risk of transferring SARS-CoV-2 from an organ donor to the recipient. Limiting such risks may be accomplished in several ways:

Deferral of nonurgent transplantations;
Pretransplant screening;
Prevention in the kidney recipients;
Vaccination.

(a) Deferral of nonurgent transplantation To minimize the risk of infection, elective transplantation, such as living-donor kidney transplantation and nonurgent transplantation, deceased donor transplantation should be avoided principally in transplant centers where the resources to treat infected kidney transplant patients are limited [53].

(b) Donor screening All donors should be screened for COVID-19 [54]. In addition, counseling adopted from the American Society of Transplantation guidelines [55] should be applied as follows:

- Decline donors with suspected COVID-19 symptoms or chest imaging in suspected positive cases;
- Decline donors with suspected COVID-19 within the past 21 days;

- Decline donors who recently had contact with persons with known COVID-19;
- Test all donors for SARS-CoV-2 infection by real-time polymerase chain reaction (RT-PCR) performed on respiratory tract samples.

For living kidney donors who test positive for SARS-CoV-2, the AST suggests waiting at least 21 days from the time of resolution of symptoms and PCR negativization before proceeding with transplantation.

(c)     Prevention in the kidney recipients Preventive measures for kidney transplant recipients are similar to those adopted for the general population (social distancing, careful hand washing, and respiratory hygiene). Additionally, as KTx recipients who have COVID-19 may shed greater amounts of virus for a longer duration [56], prolonged isolation and/or testing may be needed.

(d)     Vaccine Several studies have documented that in KTx patients the seroconversion is low in comparison to healthy controls after two doses of several different types of vaccines with an inactivated virus and of mRNA vaccines [57–59]. A recent case-controlled study [60] confirmed these data. The authors highlighted that this low efficacy is related to immunosuppression, and even by increasing the two doses to a third booster dose, COVID-19 remains a fatal disease despite different treatment modalities. According to this study, COVID-19 vaccines cannot prevent death in all KTx patients, even if they can decrease hospitalization rates and disease duration in most patients. In a very recent review on COVID-19 vaccine efficacy and immunogenicity in KTx recipients [61], the authors found that KTx patients have a decreased antibody response to COVID-19 vaccines, but third and fourth doses have shown to increase the antibody production. Based on the available data, professionals who treat KTx patients advocate for a complete four-dose vaccination against SARS-CoV-2.

*5.2. Therapeutic Strategies*

Although guidelines on the management of the SARS-CoV-2 infection are currently available for the general population, to date, guidelines regarding the management of kidney transplant patients with COVID-19 infection do not exist.

Treatment strategies have varied across centers and countries [62–64].

Table 3 shows the available treatments in COVID-19-infected patients.

**Table 3.** Therapeutic strategies for COVID-19 kidney transplant patients.

| |
|---|
| • **Modulation of Immunosuppressive Drugs** |
| • **Corticosteroids** |
| • **Biological Agents** |
| • **Intravenous Immunoglobulins** |
| • **Hyperimmune Plasma** |
| • **Antivirals** |

(a)     Modification in immunosuppression

Managing immunosuppression in a KTx recipient in the context of a severe infection is discussed. The immunosuppressed patient has a reduced immunological response. As a consequence, there is a rationale to reduce and even withdraw immunosuppression in the case of severe COVID-19 infection.

The antimetabolites should be discontinued first because of their effect on inhibiting T-cell function [65].

Discontinuation of mTOR inhibitors is discussed because they have an antiviral potential [66]. However, due to the fact that they have been associated with various types of lung injury [67], their discontinuation should also be considered.

Regarding the use of calcineurin inhibitors (CNIs), the most common approach is to minimize doses [68].

In the aforementioned systematic review of COVID-19 infection in kidney transplant patients [10], total immunosuppression was reduced in 27% and withdrawn in 31% of patients.

Antimetabolite discontinuation was as high as 91%. mTOR withdrawal was 67%.

CNIs were withdrawn in 58% and reduced in 32% of patients.

Interesting data were reported by the TANGO study [40], and the outcomes and management of hospitalized kidney transplant recipients with COVID-19.

In the study, reporting the data of 144 patients, among whom were 98 survivors and 46 nonsurvivors, the authors observed the following:

The factor principally influencing a worse outcome was significantly the need for intubation, while the presence of AKI did not influence the outcome.

Observing the management of the immunosuppression, neither tacrolimus withdrawal, MMF, or everolimus withdrawal or the increase in steroid dosage had a significant influence on the mortality rate.

In this study, only the use of antibiotics had a beneficial effect on the outcomes.

The use of different antivirals in different combinations did not have any effect. It should be highlighted that the use of antivirals (remdesivir included) has been very low in this study for an effective statistical evaluation.

(b) Corticosteroids

Dexamethasone is the only drug that demonstrated a clear survival benefit in the general population. Despite the fact that transplant patients were not included in any randomized trial, including RECOVERY trial [69], it is now standard of care to use high-dose glucocorticosteroids in transplant patients with COVID-19 pneumonia.

(c) Biologic agents

Tocilizumab, a recombinant humanized anti-interleukin-6 (IL-6) receptor monoclonal antibody is used in the treatment of severe cytokine release syndrome. In the already cited review [10], 26% of patients received tocilizumab. Tocilizumab effects, when given to such patients, are good.

Anankira, a recombinant antagonist of the IL-1 receptor, reduces cytokines involved in the inflammatory response. To date, no trial studying Anankira included kidney transplant patients [70]. Further sudies are needed to establish the Anankira effectiveness.

Less frequently applied agents include clazakizumab and leronlimab. The latter one is an agent against the CC chemokine receptor type 5 (CCR5). Unfortunately, due to the small numbers of patients enrolled with the last two drugs, it is impossible to draw conclusions about their real impact on the disease outcomes.

(d) Intravenous Immunoglobulins

High-dose intravenous immunoglobulins (IVIGs) have been proposed in patients with COVID-19 and deteriorating conditions to act against the systemic inflammatory response and the endothelial activation [71], but only a few cases have been reported to evaluate their efficacy. Overall, IVIG efficacy seems to be linked to the timing of administration. Patients might not receive much benefit when systemic damage has already taken place. Regardless, a randomized controlled trial evaluating the high-dose IVIG therapy in severe COVID-19 has been initiated to provide more evidence of such treatment.

(e) Hyperimmune Plasma

Convalescent plasma used in nontransplant patients with severe COVID-19 does not seem to improve survival rates [72]. In addition, its role in kidney transplant patients has not been evaluated.

(f)  Antivirals

Antiviral treatment is controversial since sufficient evidence to prove its efficacy is not available [73,74]. Antiviral drugs include antimalarial drugs (chloroquine and hydroxychloroquine), HIV protease inhibitors (lopinavir/ritonavir), (daranovir/cobicistat), and daranovir/ritonavir); and remdesivir [75]. Remdesivir is an inhibitor of the viral RNA-dependent RNA polymerase with an in vitro activity against SARS-CoV-2.

Its efficacy in the general population has been validated by a double-blind placebo-controlled trial including more than 1000 patients [76]. Its use is contraindicated in patients with an estimated glomerular filtration rate (eGFR) <30 mL/min, and the drug should be administered early after the infection onset to reduce the cytokine storm phase [75]. However, it should again be highlighted that to date there are no effectiveness or safety studies on the use of remdesivir in the transplanted population [77].

## 6. Conclusions and Final Comments

Many points remain to be clarified regarding COVID-19 infection involving KTx patients. The epidemiology is to date clear as well as the higher incidence of COVID-19 among dialysis patients than KTx patients. In addition, risk factors for a worse outcome have been clarified.

Main histopathologigal features are FSGS and those aspects related to acute rejection. Clinically, the most frequent aspect is AKI, often related to acute tubular necrosis. To date, it has not been definively clarified whether the kidney lesions are related to the RNA virus "per se" or to other factors, such as cytokine storm.

The prophylaxis has been well-clarified by the American Society for Organ Transplantation.

The use of vaccines to prevent the disease is mandatory considering the immuno-suppressive status of such patients. How many vaccine administrations is still doubtful, and is the subject of research. The four-dose administration is to date considered the optimum from many countries, but the future is open to different strategies. Similarly, the therapeutical strategy is open to new research. To date, the immunosuppressive reduction is mandatory, but many of the drugs used to date are still the subject of randomized trials. The use of Tocilizumab, an anti-IL-6R antibody, is largely used in the case of cytokine storm. Other anti-IL-6 are still the subject of randomized clinical trials. Among the antiviral agents, remdesivir is the most commonly used, but its efficacy has been validated only in the general population.

Finally, a comment on the references collected. The majority refers to recent years, 2022 included. It should be highlighted that this field is completely open to new studies, and while I am writing, some parts of the study run the risk of becoming "old" in a short time.

**Author Contributions:** M.S. and A.T. contributed equally to the manuscript; M.S. designed the study, performed the last revision, and provided answers to the reviewers. A.T. collected the data from literature; M.S. and A.T. analyzed the collected data and wrote the manuscript. All authors have read and agreed to the published version of the manuscript.

**Funding:** This research received no external funding.

**Institutional Review Board Statement:** Not applicable.

**Informed Consent Statement:** Not applicable.

**Data Availability Statement:** Not applicable.

**Conflicts of Interest:** The authors do not have any conflict of interest in relation to the manuscript.

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
