# Peer review of "COVID-19 and Kidney Transplantation: Epidemiology, Histopathological Presentation, Clinical Presentation and Outcomes, and Therapeutic Strategies"

_2673-3943, doi:10.3390/transplantology3030023_

Round 1

Reviewer 1 Report

 An extensive literature review covers epidemiology, pathology, clinical presentation, treatment and outcomes of COVID-19 in kidney transplant patients. This review will be of value for clinitians and researchers.

A literature review is really comprehensive and valuable. I would suggest to shortly summarise a literature data at the end of an article.

Some minor corrections of English language is needed. Sentence in lines 200-201 should be revised as the meening is not clear. Mistake in „COVID-19“ in line 222.

Author Response

ANSWER Rev 1

An extensive literature review covers epidemiology, pathology, clinical presentation, treatment and outcomes of COVID-19 in kidney transplant patients. This review will be of value for clinicians and researchers.

Thank for your comments. Indeed, our aim was to cover the latest findings in literature in the fields described

A literature review is really comprehensive and valuable. I would suggest to shortly summarize a literature data at the end of an article.

We have summarized the literature data together with the conclusions. Please look at the latest added chapter by the end of the review

Some minor corrections of English language is needed. Sentence in lines 200-201 should be revised as the meaning is not clear. Mistake in „COVID-19“ in line 222.

English writing in lines 200-201 has been rewritten, as well the COVID-19 in line 222

All the corrections have been in red

Reviewer 2 Report

The manuscript is a review, that does not add original information, but an analyzed recompilation of previous publications. 

There are typing mistakes to be corrected.

In "treatment", they should show in separate the prophylaxis and screening from the therapeutic strategies themselves. 

What does "full vaccination " means for KTx patients??

They must describe the results (good or bad) of biologic agents as therapeutic agents

The efficacy of high dose IVIGs must be described

Phrases must be connected , to read it more fluently.

A conclusion final item must be stated

Tables, since they are not own data, must mention the source origin (reference) in the table description.  

Author Response

ANSWER Rev 2

There are typing mistakes to be corrected

Typing mistakes have been corrected

In “treatment” they should show in separate the prophylaxis and screening from the therapeutic strategies themselves

The chapter has been partially rewritten according the comment

What does “full vaccination” means for KTx patients ?

To date “full vaccination” means four administrations of vaccine in most countries

They must describe the results (good or bad) of biological agents as therapeutic agents

This has been added to the main text, wherever possible

The efficacy of high dose IVIGs must be described

IVIGs efficacy has been added

Phrases must be connected, to read it more fluently

Many phrases have been connected as suggested

A conclusion final item must be stated

A conclusion chapter has been added

Tables, since they are not own data, must mention the source origin (reference) in the table description

References to the tables has been added with the exception of tables 3 and 5, because I dis them